# Impact of Ionic Liquid Structure and Loading on Gas Sorption and Permeation for ZIF-8-Based Composites and Mixed Matrix Membranes

**DOI:** 10.3390/membranes12010013

**Published:** 2021-12-23

**Authors:** Paloma Ortiz-Albo, Tiago J. Ferreira, Carla F. Martins, Vitor Alves, Isabel A. A. C. Esteves, Luís Cunha-Silva, Izumi Kumakiri, João Crespo, Luísa A. Neves

**Affiliations:** 1LAQV/REQUIMTE, Department of Chemistry, NOVA School of Science and Technology, FCT NOVA, Universidade NOVA de Lisboa, 2829-516 Caparica, Portugal; o.albo@campus.fct.unl.pt (P.O.-A.); tjo.ferreira@campus.fct.unl.pt (T.J.F.); i.esteves@fct.unl.pt (I.A.A.C.E.); jgc@fct.unl.pt (J.C.); 2Low Carbon & Resource Efficiency, R&Di, Instituto de Soldadura e Qualidade, Av. Prof. Cavaco Silva 33, 2740-120 Oeiras, Portugal; cfmartins@isq.pt; 3LEAF—Linking Landscape, Environment, Agriculture and Food—Research Center, Associated Laboratory TERRA, Instituto Superior de Agronomia, Universidade de Lisboa, Tapada da Ajuda, 1349-017 Lisboa, Portugal; vitoralves@isa.utl.pt; 4LAQV/REQUIMTE, Department of Chemistry and Biochemistry, Faculty of Sciences, University of Porto, 4169-007 Porto, Portugal; l.cunha.silva@fc.up.pt; 5Graduate School of Sciences and Technology for Innovation, Yamaguchi University, Ube 7558611, Japan; izumi.k@yamaguchi-u.ac.jp

**Keywords:** CO_2_ separation, mixed matrix membrane (MMM), metal–organic framework (MOF), ionic liquid (IL), IL@MOF composite

## Abstract

Carbon dioxide (CO_2_) capture has become of great importance for industrial processes due to the adverse environmental effects of gas emissions. Mixed matrix membranes (MMMs) have been studied as an alternative to traditional technologies, especially due to their potential to overcome the practical limitations of conventional polymeric and inorganic membranes. In this work, the effect of using different ionic liquids (ILs) with the stable metal–organic framework (MOF) ZIF-8 was evaluated. Several IL@ZIF-8 composites and IL@ZIF-8 MMMs were prepared to improve the selective CO_2_ sorption and permeation over other gases such as methane (CH_4_) and nitrogen (N_2_). Different ILs and two distinct loadings were prepared to study not only the effect of IL concentration, but also the impact of the IL structure and affinity towards a specific gas mixture separation. Single gas sorption studies showed an improvement in CO_2_/CH_4_ and CO_2_/N_2_ selectivities, compared with the ones for the pristine ZIF-8, increasing with IL loading. In addition, the prepared IL@ZIF-8 MMMs showed improved CO_2_ selective behavior and mechanical strength with respect to ZIF-8 MMMs, with a strong dependence on the intrinsic IL CO_2_ selectivity. Therefore, the selection of high affinity ILs can lead to the improvement of CO_2_ selective separation for IL@ZIF-8 MMMs.

## 1. Introduction

Membrane-based carbon dioxide (CO_2_) separation has been investigated as a more environmentally friendly technology than the traditional technology used in industry, which consists of the use of aqueous alkanolamine solutions [1]. Membranes of polymeric and inorganic natures have been studied for CO_2_ separation processes, but both types of membranes present some practical drawbacks. Polymeric membranes present a trade-off limitation of selectivity–permeability and long-term stability concerns, while inorganic membranes can overcome the Robeson trade-off upper bound, although they require a high investment cost in terms of preparation and their scale-up is difficult due to their mechanical fragility. The concept of mixed matrix membranes (MMMs) emerged as a consequence of seeking the beneficial combination of polymers with uniformly dispersed fillers (either organic or inorganic) [2,3,4,5]. However, poor filler–polymer interactions and affinity can lead to the appearance of non-ideal structures, including voids, pore blockage, polymer rigidification, or particle agglomeration, which consequently reduces membrane selectivity. Due to their hybrid nature, metal–organic frameworks (MOFs) have risen as versatile alternative materials over pure inorganic fillers. MOFs consist of metal ions/clusters coordinated by organic ligands (linkers) that can improve compatibility and interfacial interactions with the polymeric matrix [6,7].

Experimental gas permeation studies have shown an improvement in separation performance after the incorporation of MOFs within a polymeric matrix. However, the fabrication of defect-free MOF-MMMs has several restrictions in terms of gas separation performance enhancement and membrane brittleness [3,4,8], particularly at high loadings. The potential and advantages of using MOFs to produce MMMs has led to several research works that investigate these downsides [9]. One of the emerging solutions is the use of a compatible plasticizer to preserve polymer flexibility during membrane preparation. In particular, the use of ionic liquids (ILs) for this role has been proposed [10,11,12,13,14]. ILs are molten ionic salts formed by positive and negative ions and with high appeal in membrane technology due to their nonvolatility. Furthermore, task-specific ILs show high intrinsic CO_2_ affinity, which is strongly dependent on the anion considered [11,14,15,16].

In recent years, the incorporation of ILs in the preparation of MMMs has evolved, making it possible to differentiate and classify different strategies depending on the modified membrane component by the IL [4]. One attractive option is to upgrade the polymer by dispersing a certain amount of IL along its matrix and posteriorly add the desired filler. This approach has been typically reported with the nomenclature IL/MOF/polymer. The strategy relies on the use of an IL as a high CO_2_-selective plasticizer, resulting in a more flexible membrane with higher tensile strength [17], along with the enhancement of membrane permeability. Membrane selectivity is mainly affected by the dispersed MOF [18,19,20,21]. In this regard, most recent MMMs involve the use of polymerizable IL monomers or poly-ionic liquids (PILs), instead of conventional polymeric blends [22], to later disperse the MOF [17,23,24,25]. However, IL/MOF-based MMMs suffer a limitation of the amount of IL to incorporate, as high loading can lead to lower CO_2_ selectivity and membrane structural stability [26,27,28].

While this first method attempts a polymer upgrade, an alternative approach focuses on the modification of the filler to improve membrane selectivity, resulting in newly designed IL@MOF composite materials to be dispersed in the polymeric matrix [3]. There are two main types of synthesis routes to obtain IL@MOF materials: ionothermal synthesis and post-synthesis methods [12,29]. The ionothermal procedure involves the use of ILs as a solvent and template/structure-directing agent during the MOF synthesis. High affinity and compatibility between the IL and MOF are the main requisites for successful MOF synthesis with IL incorporated, restricting the combinations of IL and MOF to consider. On the other hand, post-synthesis methods, namely direct contact/wet impregnation, tandem post-synthetic modification (“ship-in-a-bottle”), and capillary action, can also be used to incorporate the IL into the MOF structure. These strategies considerably increase the potential combinations due to the lesser required compatibility. All the post-synthesis methods, except for direct contact/wet impregnation, have the objective of incorporating the IL inside the MOF cavities. However, the direct contact/wet impregnation method can lead to different stable incorporations for IL@MOF materials, i.e., the IL can remain on the MOF external surface; it can enter the MOF cavities or a combined/intermediate state between these last two. Their associated advantages differ depending on the final state and location of the IL in respect to the MOF structure.

An advantage of incorporating IL as a wetting external agent surrounding the filler is that this will work as an external modification of the MOF, which can improve MOF/polymer interactions [18]. The interphase filling by IL incorporation has been reported as a way to increase the membrane tensile strength and its CO_2_ selectivity [30].

The other potential location of the IL consists of its successful encapsulation inside the cavity of the MOF. This strategy originated from attempts to improve MOF sorption properties [12,13,31,32]. Target molecules for separation from CO_2_ (with a kinetic diameter of 3.3 Å) include nitrogen (N_2_; 3.64 Å) and methane (CH_4_; 3.8 Å), and the functionalization of MOFs through ILs has allowed the preparation of more selective materials in terms of preferential sorption towards CO_2_ [33,34]. Ban et al. [20] prepared [BMIM][Tf_2_N]@ZIF-8 composites via an in situ ionothermal route and later dispersed them in a polysulfone (PSf) membrane. IL@ZIF-8 MMMs were compared to other configurations such as polymer-IL blending and IL/ZIF-8 MMMs. All gas permeabilities were negatively affected for IL@ZIF-8 MMMs, which resulted in a more selective membrane compared to ZIF-8 MMMs and the other scenarios evaluated. A more recent study by Guo et al. [21] experimentally analyzed the encapsulation effect of [BMIM][PF_6_] via a wet impregnation method on the membrane performance. A higher amount of IL inside the ZIF-8 framework led to a gradual decrease in CO_2_ permeability, whereas the selectivity towards N_2_ reached a plateau. Fluorine-containing ILs have been the most used for these different approaches due to their affinity towards CO_2_, especially those including the anions bis(trifluoromethanesulfonyl)imide ([Tf_2_N]^−^) [18,20,35], tetrafluoroborate ([BF_4_]^−^) [21,35,36,37], and triflate ([OTf]^−^) [38]. While these anions have also reported interesting results, the impact of the IL properties on the separation performance has not been systematically discussed for IL@ZIF-8 MMMs.

In this work, three distinct ILs were incorporated into the ZIF-8. This is a widely studied MOF and was chosen because of its microporous nature and thermochemical stability [39]. As for the ILs, two cyano-based ILs with the dicyanamide ([DCA]^−^) and tricyanomethanide ([TCM]^−^) anions were selected, along with the fluorine-containing [Tf_2_N]^−^ one. All ILs share 1-ethyl-3-methylimidazolium [EMIM]^+^ as the cation. The cyano-based ILs were chosen due to their anion size and CO_2_ affinity [40,41,42]. Figure 1 summarizes the procedure followed in this study. IL@ZIF-8 composites were produced via wet impregnation and direct contact methods in a single step, followed by IL excess removal. Each IL was incorporated into ZIF-8 with two different loadings. Single-component gas sorption measurements were performed to determine the selectivity performance of the produced composites, specifically CO_2_ separation from CH_4_ or N_2_. The IL@ZIF-8 materials were also used as fillers for MMMs preparation at two different filler loadings. Pure gas and binary mixtures permeation experiments were performed to determine the selectivity performance of the prepared IL@ZIF-8 MMMs for the same gases used in gas sorption measurements. Furthermore, characterization of the pristine MOF, neat ILs, IL@ZIF-8 composites, and IL@ZIF-8 MMMs was extensively performed, oriented to the quantification of the IL loading and how that impacts composites and MMMs properties. This study is one of the first trials to tackle this question by performing a sequential study to explore and relate both the anion and IL loading effects on gas sorption and permeation.

## 2. Materials and Methods

### 2.1. Materials

The ILs 1-ethyl-3-methylimidazolium dicyanamide ([EMIM][DCA], >98%), 1-ethyl-3-methylimidazolium tricyanomethanide ([EMIM][TCM], 98%), and 1-ethyl-3-methylimidazolium bis(trifluoromethanesulfonyl)imide ([EMIM][Tf_2_N], 99%) were purchased from Iolitec GmbH (Heillbornn, Germany). The MOF ZIF-8 (Basolite^®^Z1200) was acquired from Sigma-Aldrich (Madrid, Spain). Ultrason S 6010 (PSf) was kindly supplied by BASF (Ludwigshafen am Rhein, Germany). Dichloromethane (DCM; Sigma Aldrich, Madrid, Spain, >99.9%) was used as a solvent. The gases used for sorption–desorption equilibrium measurements and gas permeation studies included He (>99%), N_2_ (99.99%), CH_4_ (99.999%), and CO_2_ (99.998%), all purchased from Praxair (Almada, Portugal).

### 2.2. Preparation of IL@ZIF-8 Composites

Two different IL loadings in IL@ZIF-8 composites were prepared by modifying the amount of IL in excess to be in contact with the MOF. ZIF-8 was previously degassed at 373 K for 24 h and the water content in the ILs was confirmed to be below 2wt.% prior to their use by a Karl-Fischer model 831 KF coulometer with diaphragm (Metrohm AG, Herisau, Switzerland).

For low IL loading composites, herein generically called IL@ZIF-8(low), the wet impregnation method was used. For this purpose, 1.5 mL of IL was dissolved in 2.25 mL of DCM. Then, 0.75 g of degassed ZIF-8 was brought into contact with the IL-DCM solution. For high IL loading composites, herein generically called IL@ZIF-8(high), the direct contact method was used. In this case, 0.75 g of degassed ZIF-8 was brought into contact with 3 mL of the IL. Afterwards, irrespective of the considered IL loading, the obtained mixture was continuously stirred at 300 rpm at room temperature for 24 h and finally washed with 100 mL of DCM while being filtered under vacuum. Then, after overnight drying at room temperature in a desiccator, the sample was further dried and degassed at 373 K for 3 h. Therefore, the actual amount to be in the prepared IL@ZIF-8(low) and IL@ZIF-8(high) will be required to be effectively quantified. This aspect will be addressed in the following Sections.

### 2.3. Preparation of IL@ZIF-8 MMMs

Pristine PSf membranes were prepared via the solvent evaporation method previously reported [37,38] and using DCM as a solvent. After the homogenous dissolution of the polymer in DCM, the solution was casted in a Teflon petri dish and slowly dried at room temperature. In the case of the MMMs, 10 and 30wt.% of IL@ZIF-8 filler was dispersed in DCM in a separate vial and then mixed with the PSf DCM solution. This mixture was sonicated for 1 h to ensure good filler dispersion in the polymer solution and was casted in a Teflon petri dish and slowly dried at room temperature.

### 2.4. Characterization of IL@ZIF-8 Composites and IL@ZIF-8 MMMs

Infrared spectra (IR) of the pristine ZIF-8, neat ILs, IL@ZIF-8 composites, and IL@ZIF-8 MMMs were obtained using a Fourier transform infrared (FT-IR) spectrometer Spectrum Two model (PerkinElmer, Madrid, Spain). The attenuated total reflectance (ATR) modulus was utilized, and measurements were performed at room temperature conditions, between 4000 and 400 cm^−1^ (spectral resolution of 4 cm^−1^).

An elemental analyzer Flash EA 1112 CHNS series (Thermo Finnigan, Milan, Italy) and an inductively coupled plasma atomic emission (ICP-AE) spectrometer (Jobin Yvon HORIBA, Longjumeau, France) were used for IL quantification in the IL@ZIF-8 composites through elemental mass balance.

Thermogravimetric analyses (TGA) of the pristine ZIF-8, neat ILs, IL@ZIF-8 composites, and IL@ZIF-8 MMMs were performed with a thermogravimetric analyzer Labsys EVO from SetaramKEP Technologies (Caluire et Cuire, Auvergne-Rhône-Alpes, France). The material was held in an alumina sample pan, and the analysis started from room temperature and went up to 773 K using a heating rate of 10 K/min and a 50 mL/min flow of argon (Ar).

The powder X-ray diffraction (PXRD) patterns of ZIF-8 and all IL@ZIF-8 composites were obtained with a MiniFlex II equipment (Rigaku, Tokyo, Japan) using Cu radiation and with an X-ray generator of 30 kV voltage and 15 mA current. Diffractograms were collected in the 2θ range between 2° and 50°, with a scanning speed of 0.5°/min and a step width of 0.02°.

The N_2_ sorption−desorption equilibrium at 77 K was performed on ZIF-8 and IL@ZIF-8 composites using an ASAP 2010 static volumetric apparatus (Accelerated Surface Area and Porosimetry System, Micromeritics, Norcross, GA, USA). Samples were previously degassed under vacuum at 373 K for 3–4 h before each measurement. From the obtained isotherm, textural properties such as specific total pore volume, micropore volume, and Brunauer−Emmett−Teller (BET) specific surface area of the materials were determined. Pore size distribution (PSD) of the materials was also determined using N_2_ data at 77 K.

Transmission Electron Microscope (TEM) images of ZIF-8 and IL@ZIF-8 composites were obtained with a TEM JEM-2100 (JEOL Ldt., Tokyo, Japan) microscope. Scanning Electron Microscopy (SEM) imaging of the surface and cross-section of the IL@ZIF-8 MMMs was obtained using a SEM JSM-6360LA microscope (JEOL Ldt., Tokyo, Japan). Energy dispersive X-ray spectroscopy (EDS) was used for zinc (Zn) and fluorine (F) dispersion analysis using a JED-2300 spectrometer.

Puncture test measurements were performed for all IL@ZIF-8 MMMs using a texture analyzer TA XT Plus Texture (Stable Micro Systems, Godalming, UK), equipped with a 2 mm diameter probe that perforated the membrane at a velocity of 1 mm/s. The measurements started from the moment the probe came into contact with the membrane surface. Time, distance, and the increasing applied force were registered until the membrane broke. At least three replicas were tested per membrane.

### 2.5. Gas Sorption Studies of IL@ZIF-8 Composites

Single-component CO_2_, CH_4,_ and N_2_ sorption–desorption equilibrium measurements were performed using a standard static gravimetric method, at 303.15 K between 0 and 16 bar of pressure, for ZIF-8 and all IL@ZIF-8 composites. A double-cell high-accuracy ISOSORP 2000 magnetic suspension balance (Rubotherm GmbH, Bochum, Germany) is the main feature of this gravimetric unit. The apparatus description, schematic description, and measuring protocol can be found elsewhere [32,43].

The gas adsorption capacity can be expressed in three different quantities, which are net (*q*_net_), excess (*q*_exc_), and total (*q*_t_). The net adsorbed amount is the only one that does not require the use of non-adsorbed probe molecules to determine the required reference state [44]. Still, sorption equilibria data is mostly published using *q*_t_, which allows the development of kinetic and thermodynamic models. To report adsorption data using this quantity, an estimation of the specific total pore volume is required (determined from N_2_ adsorption-desorption data at 77 K), along with the determination of the solid matrix density of the material, *ρ*_s_. This density was determined directly in the magnetic suspension balance by helium pycnometry at 333.15 K, in which He is assumed as an inert probe not adsorbed by the solid.

The *q*_net_ corresponds to the adsorbed amount experimentally obtained. It is given by the amount of adsorbate (gas) present in the cell with the adsorbent minus the amount of adsorbate present in the cell without the adsorbent and at the same pressure and temperature conditions, as shown in the following:(1)$$q_{\mathrm{net}}=\frac{m-m_{s}-m_{h}+V_{h}\;\rho_{g}}{m_{s}\;M_{w}}$$
where *m* is the mass weighted by the microbalance, *m*_s_ is the mass of degassed sample within the measurement cell, *m*_h_ and *V*_h_ are the mass and volume of the measurement cell that contribute to buoyancy effects, respectively, and *ρ*_g_ is the gas density at the pressure and temperature equilibrium conditions. The adsorbed quantities are related as follows:(2)$$q_{t}{=q}_{\mathrm{exc}}{+V}_{p}{\;\rho }_{g}{=q}_{\mathrm{net}}{+(V}_{p}{+1/\rho }_{s}{)\;\rho }_{g}$$
where *V*_p_ is the specific total pore volume [32].

The experimental sorption and desorption data points were fitted with a 4th-order polynomial [45].
(3)$$q_{t}{(p)=a\;p}^{4}{+b\;p}^{3}{+c\;p}^{2}+d\;p$$
where *a*, *b*, *c*, and *d* are the polynomial coefficient parameters and *p* is the pressure.

These fittings allowed the determination of the materials ideal CO_2_/CH_4_ and CO_2_/N_2_ selectivities, considering equimolar compositions (50:50). Ideal selectivity (*S*) was calculated, for a given total pressure, by:(4)$$S_{{\mathrm{CO}}_{2}{/\mathrm{CH}}_{4}}=\frac{{\;q}_{{t\;\mathrm{CO}}_{2}}}{{\;q}_{{t\;\mathrm{CH}}_{4}}}$$
(5)$$S_{{\mathrm{CO}}_{2}{/N}_{2}}=\frac{{\;q}_{{t\;\mathrm{CO}}_{2}}}{\begin{array}{c} q_{{t\;N}_{2}} \end{array}}$$

### 2.6. Gas Permeation Studies of IL@ZIF-8 MMMs

Pure gas permeabilities were obtained from transient permeation experiments. Figure A1a in Appendix A shows a scheme of the pure gas permeability set-up previously designed [46], consisting of a gas cell divided into two compartments of the same volume, separated uniquely by the membrane. Pressure monitoring is performed using two pressure transducers (referred as indicators in Figure A1a in Appendix A), each one connected to a compartment. The methodology of the experiments starts by purging air from inside the system with the to-be-tested gas, followed up by filling both compartments until the desired feed pressure is reached. Then the permeate side gas is purged, obtaining as a result the desired driving force for permeation. Pure gas permeability through the membrane is then calculated from the pressure variation in both cell compartments with time as follows:(6)$$P\cdot \;\frac{t}{L}=\frac{1}{\beta }\ln\left(\frac{\Delta p_{0}}{\Delta p\;}\right)$$
where *P* is the pure gas membrane permeability (m^2^/s), *t* is time (s), *p* is total pressure (bar), *L* is membrane thickness (m), and *β* is the system volumetric parameter experimentally determined (m^−1^).

Binary gas mixtures studies were performed to determine the selectivity of the membranes simulating the composition of flue gas (15% CO_2_ + 85% N_2_) and of biogas (40% CO_2_ + 60% CH_4_) streams (Figure A1b in Appendix A). Helium was selected as a sweep gas. Feed, retentate, and permeate concentrations were analyzed by gas chromatography using an Agilent gas chromatograph (GC) 7890B equipped with a thermal conductivity detector (TCD) and maintained at 473.15 K. The GC gas carrier was helium and the isothermal method with a PoraPlot U column connected to a Molsieve 5A column was used for analysis, purchased through Soquimica, Lda. (Lisbon, Portugal). The gas permeation experiments were performed at 303.15 K with 4 bar as feed pressure, and the experiments were carried out until the achievement of a steady state of permeate flow and permeate composition, with a variation of the experimental values below 10%. The permeability of each gas was calculated as follows:(7)$$j_{i}=\frac{\;F_{\mathrm{total}}^{f}\cdot \;y_{i}^{f}}{A}$$
(8)$$\Delta p_{i}=p^{f}\cdot y_{i}^{f}-p^{p}\cdot y_{i}^{p}$$
(9)$$P_{i}=\frac{j_{i}\cdot L}{\Delta p_{i}}$$
where *j_i_* is the molar flux of gas *i* (mol/m^2^/s); *F* is the gas flowrate (mol/s); *y* is the gas molar fraction; *A* is the membrane area (m^2^); ∆*p* is the driving force between feed and permeate sides (Pa); *p* is the pressure (Pa); *L* is the membrane thickness (m); and *P* is the membrane permeability towards gas *i* (mol/m/Pa/s). Upper indexes are f for feed side and p for permeate side.

Selectivity of gas *i* over *j* for pure and binary mixtures was calculated as the ratio of permeability coefficients:(10)$$\;\alpha_{ij}=\frac{P_{i}}{P_{j}}$$

## 3. Results and Discussion

### 3.1. ZIF-8, ILs, and IL@ZIF-8 Composites Characterization

Considering the wet impregnation and direct contact methods employed herein, it was important to verify if IL impregnation had occurred and quantify the IL amount impregnated. This was done for all IL@ZIF-8 composites through a set of complementary characterization techniques prior to the MMMs preparation. The prepared composite materials were firstly characterized to confirm the effective incorporation of IL, along with possible changes in the crystalline structure and morphology of the ZIF-8 after the IL impregnation protocols. For this purpose, FT-IR spectra, PXRD patterns, and TEM images (see Figure A2, Figure A3, Figure A4 and Figure A5 in Appendix B) were analyzed and compared with the ones of the pristine materials.

To qualitatively check the chemical composition of the prepared composites, FT-IR spectra of the pristine ZIF-8, neat ILs, and IL@ZIF-8 composites were obtained (Figure A2). Successful IL impregnation means that IL-related peaks must appear in the respective composite spectrum. While IL@ZIF-8(low) composites show almost negligible modifications, the IL@ZIF-8(high) ones present appreciable IL-related peaks with no changes to the ZIF-8-associated peaks or presence of remaining solvent or moisture. For the [EMIM][DCA]@ZIF-8 composites, the most significant peaks can be found between 2280–2000 cm^−1^, corresponding to anion C≡N stretching. The most significant peak found in [EMIM][TCM]@ZIF-8 composites also corresponds to the anion C≡N stretching [47], which is around 2150 cm^−1^. Finally, for [EMIM][Tf_2_N]@ZIF-8 composites, the S-N stretching peak at 1050 cm^−1^ can be found in the spectrum of the respective composite with high IL loading, along with N shuttling (below 650 cm^−1^) [48]. The results show that the ILs were successfully impregnated into ZIF-8.

PXRD patterns (Figure A3) were obtained for ZIF-8 and the produced IL@ZIF-8 composites. The composites’ patterns reveal peaks in the same position as the ZIF-8 one, which means that the crystalline structure after DCM usage and IL impregnation was maintained. In terms of peak intensity, [EMIM][TCM]@ZIF-8(high) has changes in peak intensity that suggest electron density changes in ZIF-8 after IL impregnation [20,31]. The same kind of diffractogram has been reported already in a study where the IL was found outside the ZIF-8 structure [49], which was related to a potential core–shell composite.

TEM images of pristine ZIF-8 and IL@ZIF-8 composites (Figure A4 and Figure A5) were also obtained. The micrographs confirm that no relevant morphological changes occurred after IL impregnation but cannot be used to confirm the presence of impregnated IL.

After confirming that the IL impregnation protocols were successful and did not damage the ZIF-8 structure, quantification techniques such as CHNS elemental analysis and ICP-AE spectroscopy for zinc (Zn) element were combined for an IL estimative quantification (see Table A1 of Appendix B). The equations utilized for the elemental mass balance were gathered in Appendix B (Equations (A1)–(A3)). Note that ZIF-8 has a distinct Zn element in its structure and sulfur (S) is only found in [EMIM][Tf_2_N]. The [EMIM][DCA] and [EMIM][TCM] loadings were estimated considering an ideal Zn-, C-, and N-based elemental balance. The average values and standard deviation were calculated from the difference between the amount calculated with several elements (e.g., C-based balance against N-based balance results). The estimated IL loadings are found in Table 1. The wet impregnation method only managed ~3wt.% IL loading for all IL@ZIF-8(low) composites. In contrast, the direct contact method provided ~30wt.% IL loading for two IL@ZIF-8(high) composites. The IL loading for [EMIM][DCA]@ZIF-8(high) is one order of magnitude below the other two. The authors cannot explain this but can confirm that the composite was produced several times following the same impregnation protocol to discard experimental errors. For this sample, a ~30wt.% loading could never be achieved under these preparation conditions.

Additionally, N_2_ sorption-desorption isotherms at 77 K were obtained (see Figure A6 of Appendix B) to determine the IL impact on the textural properties of the IL@ZIF-8 composites. Table 1 gathers the textural properties of these materials, along with pristine ZIF-8. Total pore volume (*V*_p_), specific BET surface area (*A*_BET_), and micropore volume (*V*_micro_, calculated using the Dubinin–Astakhov equation [50]) were determined for a relative pressure (*p*/*p*_0_) of 0.97. The textural properties of IL@ZIF-8(low) composites are very similar, though slightly inferior, to the ZIF-8 ones. Therefore, N_2_ data at 77 K is in accordance with the very low IL loadings quantitatively determined. As for the IL@ZIF-8(high) composites, it is relevant to observe the textural properties of [EMIM][TCM]@ZIF-8(high) and [EMIM][Tf_2_N]@ZIF-8(high), as they were the only composites that did not show a Type I isotherm, according to IUPAC classification [51], typical of microporous materials. Both total pore volume and specific BET surface area values indicate that these two composites are nonporous materials. This means that the IL either occupied all the pore volume or simply blocked it. Combining both N_2_ sorption-desorption at 77 K and PXRD data for [EMIM][TCM]@ZIF-8(high), the IL potentially stayed mostly outside the porous structure of the MOF, thus completely blocking it. On the same data combination for [EMIM][Tf_2_N]@ZIF-8(high), the PXRD diffractogram was very similar to the ZIF-8 one, suggesting another potential arrangement. The IL was able to fill the porous volume of ZIF-8 and, with no more space to fill, it stayed outside of the MOF structure. According to a study found in the literature [49], the difference in the nature of the IL and MOF might play a role in preventing a hydrophilic IL (as [EMIM][TCM]) with very high loading from entering the structure of a hydrophobic MOF (and vice versa).

Using non-local density functional theory (NLDFT) calculations combined with the N_2_ data at 77 K, the pore size distribution (PSD) of ZIF-8 and the produced IL@ZIF-8 composites were obtained (Figure A7 of Appendix B), confirming that ZIF-8 and the porous composites are microporous materials.

TGA profiles were obtained for pristine ZIF-8, neat ILs, and all IL@ZIF-8 composites to analyze the impact of IL impregnation on the thermal stability of the materials (Figure 2). IL@ZIF-8(low) composites presented TGA profiles closer to the pristine MOF, and IL@ZIF-8(high) composites, with the exception of [EMIM][DCA], showed profiles closer to the neat IL. Results suggest an initial step on the degradation caused by remaining moisture and the IL, which possesses lower thermal stability compared to ZIF-8. It is followed by the conjunct degradation of the remaining IL and ZIF-8. The differences on the starting degradation temperature between the IL and respective composites seem to be dependent on the IL nature. For instance, [EMIM][TCM]@ZIF-8(high) affected the rate of degradation compared to pristine IL and ZIF-8, decomposing faster than its IL bulk phase as a consequence of IL–MOF interactions. Conversely, [EMIM][Tf_2_N]-based materials degraded similarly, regardless of being impregnated or in its bulk phase. From the remaining sample mass at the end of the assays, TGA profiles also validate the order of magnitude of the estimated IL loadings.

### 3.2. Sorption Performance of the IL@ZIF-8 Composites

Figure 3 shows the single-component CO_2_, CH_4,_ and N_2_ sorption-desorption isotherms that were obtained for ZIF-8 and the IL@ZIF-8 composites. No hysteresis was found in any of the materials, showing complete reversible behavior and full regeneration after the sorption of the studied gases. All the sorption-desorption isotherms are Type I, according to the IUPAC classification [51], which is typical of microporous materials.

In the pressure range studied, ZIF-8 showed a superior gas uptake than all the IL@ZIF-8 composites, denoting the physisorption character of all the selected ILs [43]. For the IL@ZIF-8(low) composites, considering they have similar estimated mass loadings, CO_2_ sorption presented a trend depending on the anion ([DCA]^−^ < [TCM]^−^ < [Tf_2_N]^−^), which is consistent with the CO_2_ Henry constants at 303.15 K for [EMIM][DCA] (78.00 bar), [EMIM][TCM] (61.43 bar), and [EMIM][Tf_2_N] (36.50 bar) [52,53]. For the IL@ZIF-8(high) composites, [EMIM][DCA]@ZIF-8(high) sorbed less CH_4_ and N_2_ than [EMIM][DCA]@ZIF-8(low). However, for CO_2_, both composites present a similar behavior. Therefore, the amount of IL in [EMIM]DCA]@ZIF-8(high) somehow compensated the loss of textural properties and provided higher ideal selectivity values. However, the most noteworthy results from sorption-desorption measurements were of the [EMIM][TCM]@ZIF-8(high) and [EMIM][Tf_2_N]@ZIF-8(high) composites. Both materials showed negligible textural properties, yet the former sorbed very little gas while the latter sorbed significantly more. For [EMIM][TCM]@ZIF-8(high), it is shown that a core–shell composite was obtained [49], with the gas solubilizing in the IL and not being allowed into the porous volume of the ZIF-8. In contrast, as [EMIM][Tf_2_N] presents a higher affinity towards CO_2_ (almost half the CO_2_ Henry constant), the composite showed some sorption for the three gases.

Figure 4 shows the calculated ideal CO_2_/CH_4_ and CO_2_/N_2_ selectivities for ZIF-8 and the IL@ZIF-8 composites, using Equations (4) and (5) and considering equimolar mixtures (50:50). The obtained coefficient parameters, along with the global average relative error (ARE) of the fitting, can be found in Table A2, Table A3 and Table A4 of Appendix B. For both mixtures, all composites improve the selectivity performance up until 4 bar. [EMIM][TCM]@ZIF-8(high) and [EMIM][Tf_2_N]@ZIF-8(high) presented the highest increase in selectivity when compared to ZIF-8. A note should be made here for [EMIM][TCM]@ZIF-8(high) composite, as there is a clear gas uptake–selectivity trade-off when incorporating IL into the MOF. However, this material sorbs very low gas quantities. This means that, to use this material in a gas separation process, a lot more of this material would be required when compared with other adsorbent materials that have superior gas uptake albeit inferior selectivity. Finally, as previously mentioned, [EMIM][DCA]@ZIF-8(high) showed superior selectivity performance than [EMIM][DCA]@ZIF-8(low) in the entire pressure range studied.

### 3.3. IL@ZIF-8 MMMs Characterization

After preparation of the MMMs through the solvent evaporation method, several techniques for the characterization of their chemical composition and thermal and mechanical stabilities were employed. Figure 5 gathers cross-section SEM images of several [EMIM][Tf_2_N]-based MMMs (other membranes and surface images can be found in Figure A8 in Appendix C). It should be mentioned that a more complete analysis was made for the [EMIM][Tf_2_N] IL, as it is the IL that allows a comparison between the elemental dispersion of ZIF-8 (Zn element) and IL (F element). Therefore, 10wt.%[EMIM][Tf_2_N]@ZIF-8(low)(PSf), 10wt.%[EMIM][Tf_2_N]@ZIF-8(high)(PSf), 30wt.%[EMIM][Tf_2_N]@ZIF-8(low)(PSf), and 30wt.%[EMIM][Tf_2_N]@ZIF-8(high) (PSf) were accompanied by EDS analysis mapping for Zn and F distribution characterization (Figure 5b,d,f,h).

The prepared membranes with different loadings of the composites were found to be dense with non-apparent defects or voids. For MMMs, cross-section images presented a more heterogeneous granulated-like texture due to the presence of ZIF-8 or IL@ZIF-8 composites, which increased with higher loading. EDS mapping images also showed a good dispersion of fillers along the cross-section images. The elements F and Zn were distributed uniformly along the cross-section with no agglomerations, regardless of the loading considered.

Figure A9 in Appendix C shows the normalized FT-IR spectra of the PSf-based MMMs for the analysis of their chemical composition. PSf membrane FT-IR spectrum was characterized by the presence of 1328 and 1157 cm^−1^ for SO_2_ asymmetric and symmetric bonds. Aromatic C=C, C-S-C, C-O-C, and aliphatic C-H were also found at 1593, 715 and 695, 1259, and 2970 cm^−1^, respectively. These peaks were persistently found in all the prepared membranes [28,54,55]. In Figure A9a, MMMs composed only of ZIF-8 as filler are included for two loadings: 10 and 30wt.%. The presence of ZIF-8 was confirmed with one of its most representative peaks at 1584 cm^−1^ (C=N stretching) and other bands in 600–1500 cm^−1^ (bending modes of imidazole linker) [56,57,58]. The mentioned peaks and others in the range of 1350 to 900 cm^−1^, associated to the in-plane bending of the imidazolium ring, were visualized with increasing magnitude along with the filler loading, while PSf peaks were proportionally reduced.

Similarly, 10wt.% composite-MMMs (Figure A9b–d) presented some of the characteristic peaks, whose intensity increased for 30wt.%[EMIM][Tf_2_N]@ZIF-8 (PSf) (Figure A9e). As a consequence of the low IL concentration, the presence of IL was not detected by FT-IR spectra. Band shifts or new bonds were not observed, therefore being mostly a physical interaction and blend of the components, without chemical interaction or linking between polymer and filler.

Figure 6 shows the thermal decomposition until 773 K of prepared MMMs with low IL-based composites (Figure 6a); high IL-based composites (Figure 6b); and higher composite loading in MMMs (Figure 6c), compared to pristine PSf and ZIF-8-based MMMs. Table 2 gathers the estimated decomposition temperature of the membranes tested.

The PSf membrane remained stable until reaching its decomposition temperature of 749 K (Table 2) [59]. The membranes abruptly started to decompose, reaching below 90wt.% of initial mass sample at 773 K. Similarly, the 10wt.% ZIF-8 (PSf) membrane showed signs of decay sooner, with an intermediate decomposition of less than 5wt.% at 473 K, which is potentially related to moisture in the membrane. The higher loading of ZIF-8 (see Figure 6c) effectively improved the thermal stability, compared to the PSf membrane, with a final mass of 95wt.% at 773 K and a not significant variation of the thermal decomposition temperature for this loading.

As a general comment, the rest of the prepared membranes did not show a significant amount of remaining solvent or moisture (room temperature up to 423 K), with the main differences being observed at high temperatures, and more specifically, at their decomposition temperature (Table 2).

Regarding the IL impact on thermal stability, a low amount of IL (Figure 6a) did not show apparent deviation compared to ZIF-8-based MMMs, except for the [EMIM][DCA] composite, as its thermal decomposition was the most affected compared to pristine ZIF-8 with a decrease from an average of 753 K for the other membranes to 687 K in the decomposition. On the other hand, for the IL@ZIF-8(high) composites, mass decay started at around 423 K for [EMIM][TCM]; 523 K for [EMIM][DCA]; and [EMIM][Tf_2_N]-based composite MMMs at 673 K. Higher loadings of [EMIM][Tf_2_N]@ZIF-8 composites (Figure 6c) improved the thermal stability for low amounts of IL incorporation, while higher loadings did not significantly improve the thermal behavior regardless of the ZIF-8 loading increase, being approximately 683 K for its decomposition temperature regardless of IL loading.

However, the incorporation of IL did not significantly affect the thermal behavior of MMMs compared to pristine PSf, being the most thermically favorable membranes prepared with [EMIM][Tf_2_N] at higher loadings.

Puncture tests were performed for all the prepared membranes and their normalized tensile strengths (divided by membrane thickness) were gathered in Table 2. Values and deviation were calculated from membrane replicas.

Incorporation of ZIF-8 in the PSf matrix decreased the membrane strength substantially upon puncture for 10wt.% and 30wt.% of ZIF-8. This fact is attributed to the presence of ZIF-8 crystals, which create discontinuity zones within the dense polymeric PSf. This behavior was not observed for the prepared MMMs, as the puncture stress values were closer to those of PSf, especially for IL@ZIF-8(high) samples. The presence of ILs made the membranes less brittle, making them more resistant upon perforation. The IL is likely to be around ZIF-8 particles, with the roles of wetting agent and gap filling in the polymer/ZIF-8 system [18].

### 3.4. Gas Permeation

#### 3.4.1. Ionic Liquid Influence—Anion and Loading

CO_2_, CH_4,_ and N_2_ pure gas permeation tests at 303.15 K were performed for the prepared membranes containing a fixed amount of filler (10 wt%) and varying IL in composites ([EMIM][DCA]@ZIF-8, [EMIM][TCM]@ZIF-8, and [EMIM][Tf_2_N]@ZIF-8) and its loading (low and high) in PSf. Membrane permeability values were included comparing ideal selectivities to the Robeson upper bound (2008) at 303.15 K for CO_2_/CH_4_ and CO_2_/N_2_ [60,61] (Figure 7), and permeability experimental data can be found in Table A5 in Appendix C. CO_2_ permeability and horizontal error deviation were estimated from membrane replicas and ideal selectivity deviation from the error propagation of CO_2_ and the other corresponding gas permeability deviations.

MMMs prepared with low IL-loaded composites showed an improvement of CO_2_ permeability following the increasing CO_2_ affinity of IL anions ([DCA]^−^ < [TCM]^−^ < [Tf_2_N]^−^), which agreed with bulk IL Henry constants for CO_2_ at 303.15 K [52,53] and follow the trend obtained in Figure 3a. However, in contrast to sorption observations, the CO_2_ permeability was improved compared to ZIF-8 MMMs. This enhancement can be caused by the effect of the polymeric matrix and the synergy caused by IL-MOF-polymer at the interface. Therefore, assuming that the IL presence has an effect caused by the interaction with the polymeric matrix, the incorporated IL is likely to be at the outer external surface of ZIF-8, which due to the low amount of IL has not been confirmed and complicates its determination through a specific characterization technique.

Regarding the ideal selectivity improvement towards N_2_ (Figure 7b), a trend is presented in Figure 4b and in previous studies [46,47,62,63] of [DCA]^−^ < [Tf_2_N]^−^ < [TCM]^−^. On the other hand, a trend of [DCA]^−^ < [TCM]^−^ < [Tf_2_N]^−^ was obtained for CH_4_, presenting a similar trend to CO_2_ but in the opposite direction (Table A5 in Appendix C).

Particularly for [EMIM][DCA] composites, no apparent change in performance can be related to the loading difference. A small change in IL loading, from 3wt.% to 7wt.%, had negligible effects on the permeation.

Polysulfone membranes, as a glassy polymer, are mainly affected in the diffusion mechanism by the incorporation of MOFs, as it was described by Tanh Jeazet et al. [64]. Therefore, as similar permeabilities were obtained for high loaded composites, the affected mechanism was potentially related to a more selective material without comprising the diffusion enhancement observed in 10wt.%ZIF-8 (PSf). These observations, along with the description in Section 3.2. of non-microporous materials with low sorption at high loadings (Table 1 and Figure 3) but with outstanding sorption selectivity (Figure 4), suggest a potential incorporation of IL in the outer surface of ZIF-8 particles, as it was discussed particularly for [EMIM][TCM]@ZIF-8(high) with a potential core–shell arrangement.

Therefore, the location of IL seems to play an important role in MOF–polymer interactions, strongly affected by the nature and IL gas affinity at low IL loadings. The effect was not directly observed for single gas sorption studies for the composites. However, although experimental characterizations and gas separation results suggest a potential external presence of IL in ZIF-8 particles, no experimental technique can confirm this for low loadings of IL in MMMs.

#### 3.4.2. Composite Loading Influence in MMMs

As in the previous section, CO_2_, CH_4,_ and N_2_ pure gas permeation tests at 303.15 K were performed. From the set of results in Figure 5, [EMIM][Tf_2_N] composites were selected to analyze their impact with a higher loading (30wt.%) and compare to ZIF-8-based MMMs. Once again, membrane permeability was included comparing ideal selectivities to the Robeson upper bound (2008) at 303.15 K for CO_2_/CH_4_ and CO_2_/N_2_ [60,61] (Figure 8), and permeability experimental data can be found in Table A5 in Appendix C.

IL effect at a higher composite loading followed a similar trend compared to 10wt.% filler. Low loading of IL in composites showed a slight enhancement of CO_2_ permeability and selectivity with respect to ZIF-8 MMMs, while a higher IL content suffered an intense selectivity improvement, and therefore, CO_2_ permeability was kept similar to ZIF-8 MMMs.

However, a difference in total IL content of approximately 1 (from 30wt.% of EMIM][Tf_2_N]@ZIF-8(low)) to 9wt.% (from 30wt.% of EMIM][Tf_2_N]@ZIF-8(high)) in the membrane did not show a significant difference, as the results converged in similar permeability and selectivity values. This suggests that at higher composite values, the ZIF-8 particles start to have certain interactions and therefore, the IL potentially present in the MOF–polymer interphase will also interact with other IL@ZIF-8 composite particles. Therefore, the small differences in the total loading of IL will be of less significance in the global membrane performance.

#### 3.4.3. Gas Mixture Effect

Binary gas mixtures were tested considering 85% N_2_ + 15% CO_2_, and 40% CO_2_ + 60% CH_4_ at 303.15 K for 10wt.%[EMIM][TCM]@ZIF-8 low (PSf), 10wt.%[EMIM][Tf_2_N]@ZIF-8 (low) (PSf), 10wt.%[EMIM][TCM]@ZIF-8 (high) (PSf), 10wt.%[EMIM][Tf_2_N]@ZIF-8 (high) (PSf), and 30wt.%[EMIM][Tf_2_N]@ZIF-8 (high) (PSf) MMMs.

Membrane 30wt.%[EMIM][Tf_2_N]@ZIF-8 (low) (PSf) was not tested as it did not remain with a transmembrane total pressure of 4 bar. Lower total pressure data was not possible to obtain due to gas chromatography detection limitations. The mechanical stability here was therefore the key difference between 30wt.%[EMIM][Tf_2_N]@ZIF-8(low)(PSf) and 30wt.%[EMIM][Tf_2_N]@ZIF-8(high) (PSf), which did not differ in the preliminary tests with pure gases (Figure 8). Experiments lasted until the achievement of a steady state and stability of permeate flow and composition, analyzed by gas chromatography, with a variation of the experimental conditions below 10%. The achievement of a steady state was set when experimental parameters along with time presented a deviation below 10% from the average value. Therefore, the permeability value deviations were omitted from Table 3.

Table 3 compares the obtained membrane permeability of pure gas permeation against the permeability under binary mixture conditions.

As a general trend for [EMIM][Tf_2_N] containing MMMs, experimental CO_2_ permeabilities were similar for pure and binary mixture gas permeation tests, while CH_4_ and N_2_ permeabilities decreased in gas mixture experiments, resulting in an increased membrane selectivity towards CO_2_. This improvement was associated with the membrane preferential behavior towards CO_2_ under the presence of gas mixtures_._ This preferential sorption under mixture conditions has been previously reported by Ricci et al. (2020) [65], where it was shown that competitive sorption enhanced the solubility selectivity of glassy polymers by excluding CH_4_ over CO_2_, which led to a higher permselectivity of the materials. Additionally, all composites presented higher ideal sorption selectivity from 1.7 to 4 bar of absolute pressure (Figure 4), except for [EMIM][TCM]@ZIF-8(high).

Particularly, [EMIM][TCM] MMMs were not promoted by the competitive sorption phenomenon as the CO_2_ permeability decreased under the presence of CH_4_ or N_2_ gas. This effect was also observed for 30wt.%[EMIM][Tf_2_N]@ZIF-8(high) (PSf), although the accompanying decrease in CH_4_ permeability resulted in an increase in selectivity.

Therefore, [EMIM][Tf_2_N]@ZIF-8-based MMMs showed an important practical advantage over [EMIM][TCM] for the evaluated compositions and experimental conditions due to the preferential sorption of this IL towards CO_2_. Potential sorption studies with gas competition can optimize the screening of materials and their effect on posterior membrane CO_2_ selective behavior.

Comparison of experimental data with other works in the literature was done using an “Enhancement Filler Index” (F_index_) and by separating the membranes into two groups: IL@filler—MMMs and IL/filler—MMMs. The enhancement filler index was defined as the ratio of the MMM performance (permeability or selectivity) under the best reported conditions to a base case. For the first group, the base case was defined as the pristine polymer, while the latter was the corresponding blend without the filler.

Table A6 in Appendix C gathers the experimental works and details used for the development of Figure 9, such as the reference base utilized and the calculated data. The x-axis was defined as “Enhancement Filler Index of Membrane permeability towards CO_2_” and the y-axis as “Enhancement Filler Index of ideal selectivity”. Figure 9a corresponds to CO_2_/CH_4_ experimental data and Figure 9b, for CO_2_/N_2_, in logarithmic scale. Additionally, data from mixtures was also included for all the tested membranes and the data was connected with dashed lines.

In general terms, reported data of IL/filler/polymer showed a higher enhancement for both CO_2_ permeability and selectivity. This can be related to the differences between IL and MOF loadings which are generally higher in IL/filler/polymer. IL@ZIF-8 MMMs mostly take low loadings of filler and IL.

Regarding the prepared membranes in this work, following a similar approach, higher selectivity values were obtained with similar CO_2_ permeability enhancement.

For higher composite loadings which were less common in the literature, the CO_2_ permeability enhancement factor reached enhancement values like the membranes with the IL/filler approach, thus being enhanced for the [EMIM][Tf_2_N] case for gas mixtures.

## 4. Conclusions

CO_2_-selective composites and MMMs were prepared by incorporating ILs using the wet/direct contact method with posterior excess removal. Several techniques were employed to confirm and quantify the incorporation of IL, obtaining two groups of composites: around 3wt.% IL and 30wt.% IL in IL@ZIF-8, with the exception of [EMIM][DCA] at higher loadings.

Pure gas sorption and pure gas permeation studies showed an improvement of ideal sorption selectivity towards CO_2_, CH_4,_ and N_2_ compared to ZIF-8-based MMMs, with trends directly related to the IL intrinsic gas affinity, therefore, being one key aspect in the preparation of IL@ZIF-8 MMMs. However, gas mixture studies become essential to assure a selective behavior under gas permeation competition.

IL@ZIF-8 MMMs showed an improvement in CO_2_ permeability and selectivity against CH_4_ and N_2_ for low loading composites dispersed in the polymeric matrix. While high loading composite-based MMMs showed only a strong improvement in selectivity. At higher composite loadings for the most affine IL, the membrane performance was not significantly affected by IL loading in the composite, showing a close convergence of the MMM permeation properties. These results can be related to an improvement in the compatibility between MOF–polymer by the filling of the interphase with IL and therefore, being potentially incorporated in the external ZIF-8 area.

## Figures and Tables

**Figure 1 membranes-12-00013-f001:**
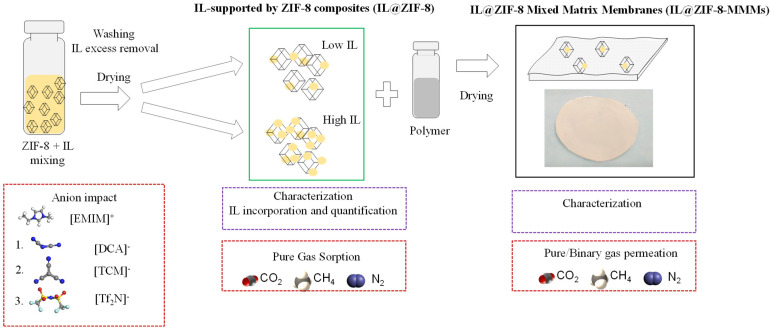
Scheme of the methodology followed in this study.

**Figure 2 membranes-12-00013-f002:**
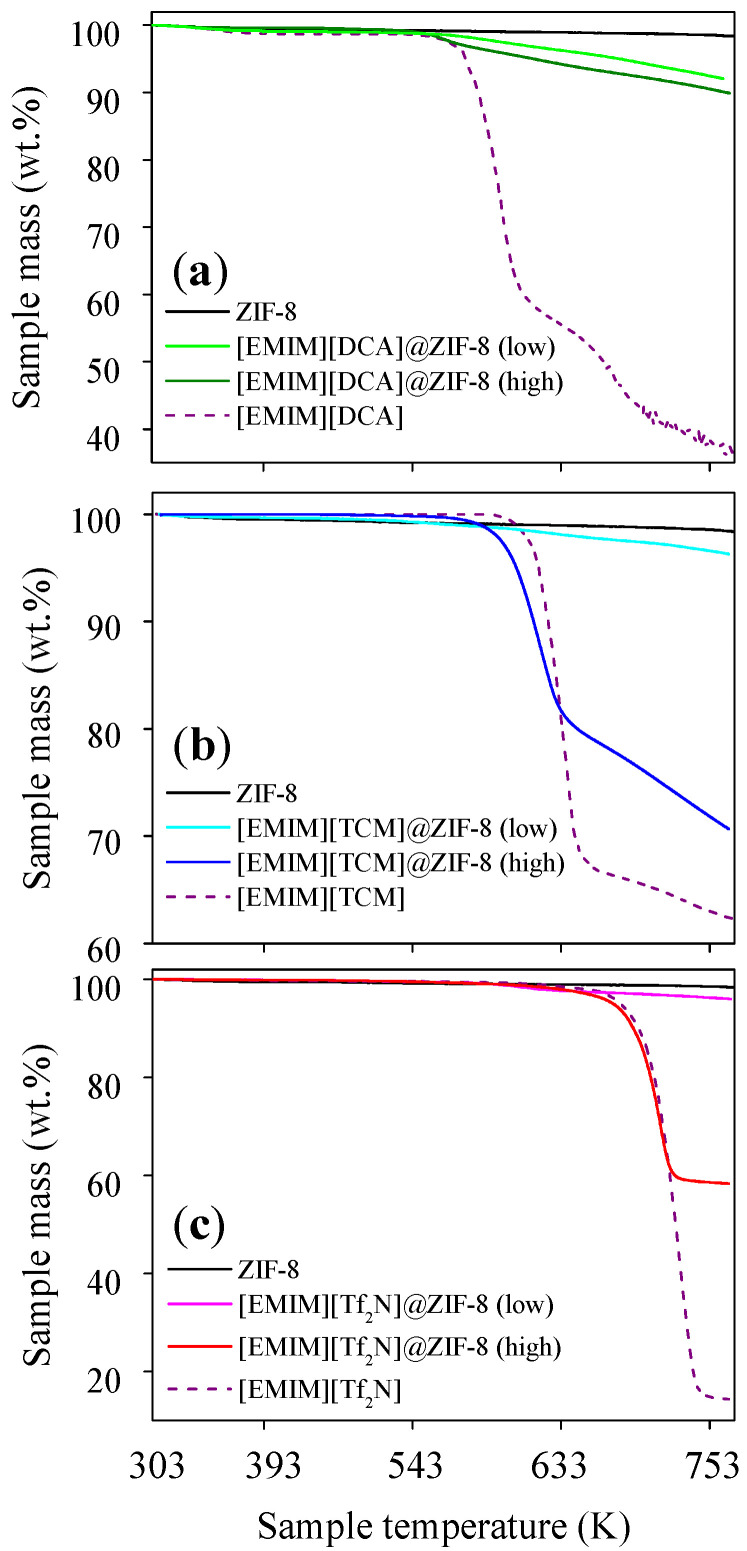
TGA profiles of pristine ZIF-8, neat ILs, and IL@ZIF-8 composites with (**a**) [EMIM][DCA]; (**b**) [EMIM][TCM]; and (**c**) [EMIM][Tf_2_N].

**Figure 3 membranes-12-00013-f003:**
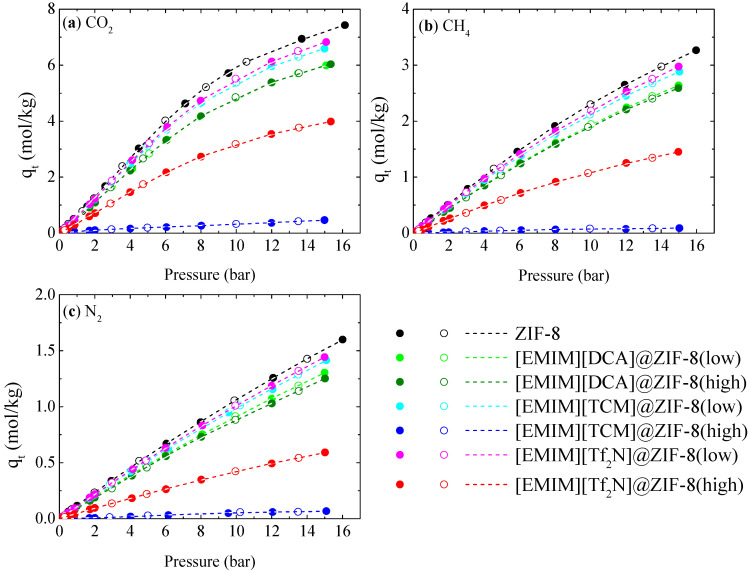
Single-component (**a**) CO_2_, (**b**) CH_4,_ and (**c**) N_2_ sorption–desorption equilibrium isotherms for ZIF-8 and IL@ZIF-8 composites at 303.15 K. Closed and open symbols denote sorption and desorption experimental data points, respectively. Dashed lines represent the 4th-order polynomial fitting to the experimental data points.

**Figure 4 membranes-12-00013-f004:**
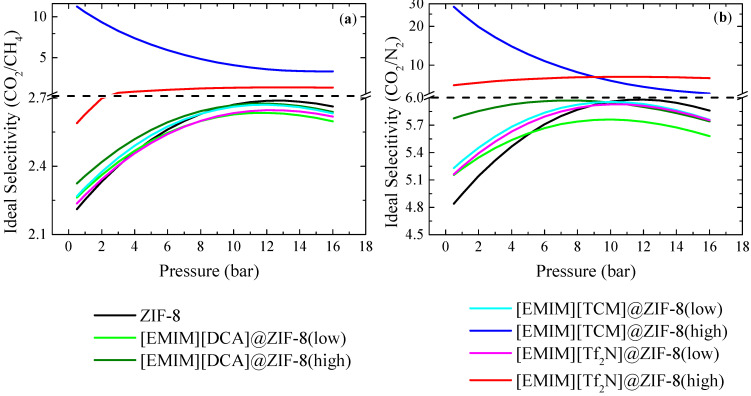
(**a**) Ideal CO_2_/CH_4_ and (**b**) CO_2_/N_2_ selectivities for ZIF-8 and the produced IL@ZIF-8 composites, considering 50:50 mixture composition.

**Figure 5 membranes-12-00013-f005:**
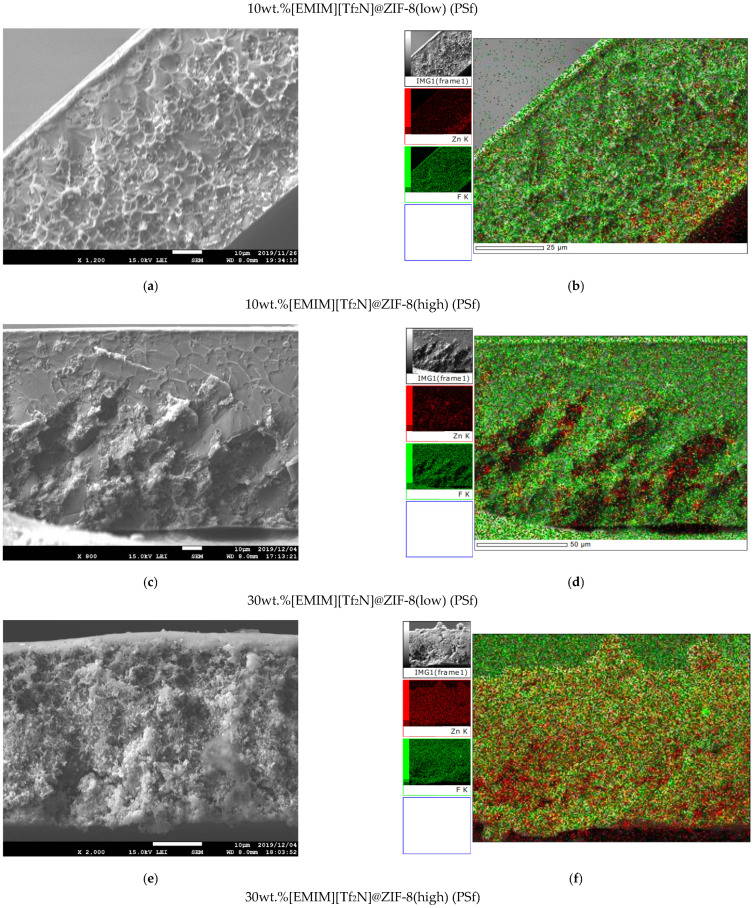
SEM images of cross-section membranes using [EMIM][Tf_2_N]@ZIF-8 composites, accompanied by EDS mapping (**a**,**b**) 10wt.%[EMIM][Tf_2_N]@ZIF-8(low)(PSf); (**c**,**d**) 10wt.%[EMIM][Tf_2_N]@ZIF-8(high) (PSf); (**e**,**f**) 30wt.%[EMIM][Tf_2_N]@ZIF-8(low)(PSf); and (**g**,**h**) 30wt.%[EMIM][Tf_2_N]@ZIF-8(high) (PSf).

**Figure 6 membranes-12-00013-f006:**
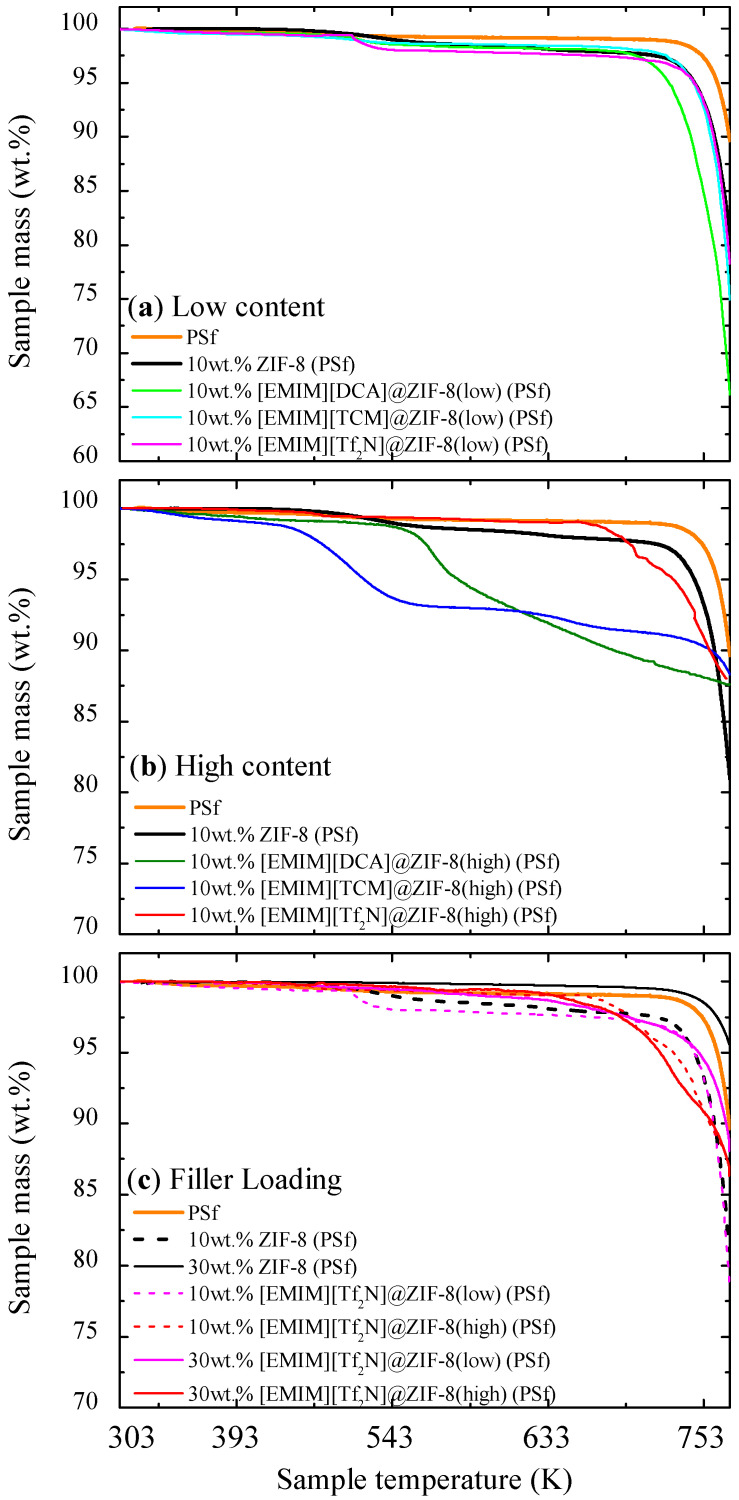
Thermal stability of PSf-based MMMs considering (**a**) 10wt.%IL@ZIF-8 composites low MMMs; (**b**) 10wt.%IL@ZIF-8 composites high MMMs; and (**c**) 10wt% and 30wt.%[EMIM][Tf_2_N]@ZIF-8 composites (low and high) MMMs.

**Figure 7 membranes-12-00013-f007:**
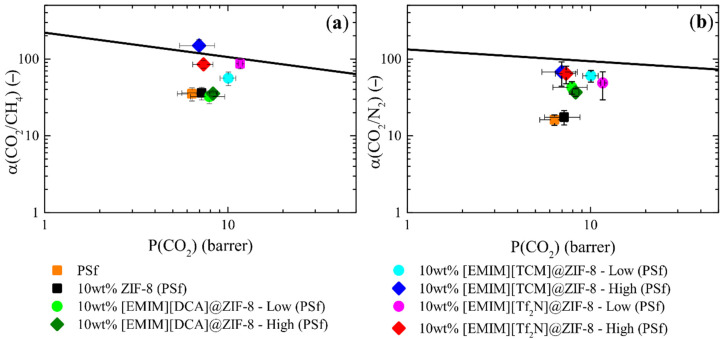
CO_2_ pure gas permeation results at 303.15 K for PSf and several MMMs with fixed 10wt.% filler compared to Robeson upper bound (2008) at 303.15 K for (**a**) CO_2_/CH_4_ and (**b**) CO_2_/N_2_ ideal selectivities.

**Figure 8 membranes-12-00013-f008:**
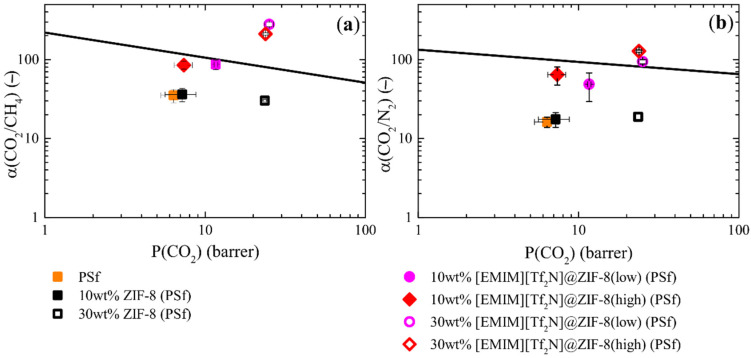
CO_2_ pure gas permeation results at 303.15 K for PSf and several MMMs with [EMIM][Tf_2_N]@ZIF-8 composites with different loadings compared to Robeson upper bound (2008) at 303.15 K for (**a**) CO_2_/CH_4_ and (**b**) CO_2_/N_2_ ideal selectivities.

**Figure 9 membranes-12-00013-f009:**
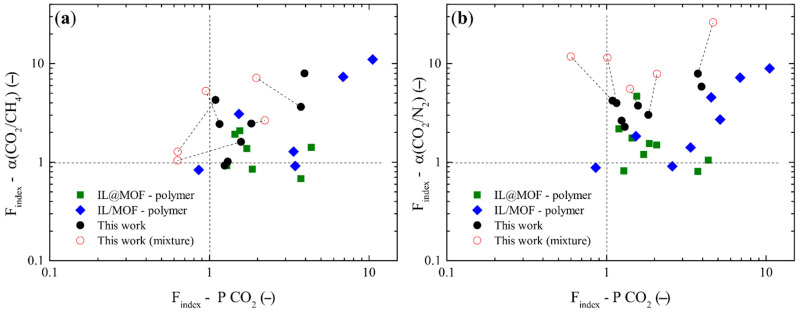
Enhancement CO_2_ permeability and selectivity of prepared membranes and reported data from Appendix C for (**a**)CO_2_/CH_4_ and (**b**) CO_2_/N_2_ separations.

**Table 1 membranes-12-00013-t001:** Estimated IL mass loading for the produced IL@ZIF-8 composites. BET surface area and porous volume obtained from N_2_ isotherms at 77 K for ZIF-8 and composites.

Sample	Estimated IL Loading (wt.%)	*V*_p_ (cm^3^/g)	*V*_micro_ (cm^3^/g)	*A*_BET_ (m^2^/g)
ZIF-8	-	0.682	0.663	1889
[EMIM][DCA]@ZIF-8(low)	3.1 ± 0.1	0.669	0.652	1851
[EMIM][DCA]@ZIF-8(high)	5.6 ± 1.3	0.552	0.533	1502
[EMIM][TCM]@ZIF-8(low)	1.0 ± 0.3	0.626	0.606	1720
[EMIM][TCM]@ZIF-8(high)	29.6 ± 0.6	0.018	0.006	9
[EMIM][Tf_2_N]@ZIF-8(low)	3.4 ± 1.3	0.658	0.640	1805
[EMIM][Tf_2_N]@ZIF-8(high)	30.5 ± 4.0	0.003	0.002	5

**Table 2 membranes-12-00013-t002:** Decomposition temperature (*T*_onset_) and normalized tensile strength of the prepared MMMs compared to pristine PSf.

Membrane	*T*_onset_ (1 Step/2 Step) (K)	Normalized Tensile Strength (MPa/mm)
PSf	749/-	64.9 ± 7.8
10wt.%ZIF-8 (PSf)	750/-	18.8 ± 1.8
10wt.%[EMIM][DCA]@ZIF-8(low)(PSf)	687/-	57.9 ± 5.8
10wt.%[EMIM][DCA]@ZIF-8(high) (PSf)	524/580	57.50 ± 2.2
10wt.%[EMIM][TCM]@ZIF-8(low)(PSf)	752/-	49.6 ± 3.4
10wt.%[EMIM][TCM]@ZIF-8(high) (PSf)	496/757	64.5 ± 3.2
10wt.%[EMIM][Tf_2_N]@ZIF-8(low)(PSf)	758/-	38.1 ± 9.0
10wt.%[EMIM][Tf_2_N]@ZIF-8(high) (PSf)	676/-	45.6 ± 6.0
30wt.%ZIF-8 (PSf)	771/-	17.2 ± 2.1
30wt.%[EMIM][Tf_2_N]@ZIF-8(low)(PSf)	684/-	12.8 ± 4.7
30wt.%[EMIM][Tf_2_N]@ZIF-8(high) (PSf)	686/-	47.7 ± 18.8

Note: “/-” denotes a single decomposition step.

**Table 3 membranes-12-00013-t003:** Gas permeability and selectivity values of IL@ZIF-8 MMMs pure gas at 303.15 K for pure gas and two binary mixtures: 85% N_2_ + 15% CO_2_ and 40% CO_2_ + 60% CH_4_.

Membrane	Composition	Permeability (Barrer)			Selectivity (-)	
		CO_2_	CH_4_	N_2_	α(CO_2_/CH_4_)	α(CO_2_/N_2_)
10wt.%[EMIM][TCM]@ZIF-8(low)(PSf)	100%	10.05	0.18	0.10	56.29	60.43
	85% N_2_ + 15% CO_2_	8.96	-	0.1	-	89.60
	40% CO_2_ + 60% CH_4_	4.02	0.11	-	36.55	-
10wt.%[EMIM][TCM]@ZIF-8(high) (PSf)	100%	6.95	0.05	0.10	149.93	67.97
	85% N_2_ + 15% CO_2_	4.03	-	0.02	-	191.00
	40% CO_2_ + 60% CH_4_	3.82	0.09	-	44.78	-
10wt.%[EMIM][Tf_2_N]@ZIF-8(low)(PSf)	100%	11.67	0.13	0.24	86.48	48.66
	85% N_2_ + 15% CO_2_	13.20	-	0.10	-	127.33
	40% CO_2_ + 60% CH_4_	14.14	0.15	-	92.84	-
10wt.%[EMIM][Tf_2_N]@ZIF-8(high) (PSf)	100%	7.36	0.09	0.11	81.78	66.91
	85% N_2_ + 15% CO_2_	6.46	-	0.03	-	184.86
	40% CO_2_ + 60% CH_4_	6.05	0.03	-	184.85	-
30wt.%[EMIM][Tf_2_N]@ZIF-8(high) (PSf)	100%	23.70	0.11	0.19	215.45	127.73
	85% N_2_ + 15% CO_2_	29.70	-	0.07	-	424.29
	40% CO_2_ + 60% CH_4_	12.52	0.05	-	250.40	-

## Data Availability

Data is contained within the article.

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
