# Peer review of "Impact of Ionic Liquid Structure and Loading on Gas Sorption and Permeation for ZIF-8-Based Composites and Mixed Matrix Membranes"

_membranes, 2021, doi:10.3390/membranes12010013_

Round 1
Reviewer 1 Report
The manuscript "Impact of ionic liquid structure and loading on gas sorption and permeation for ZIF-8-based composites and mixed matrix membranes” by Ortiz-Albo P. et al. deals with the elaboration and characterization of the composite membranes based on ZIF-8 and ionic liquids with different loadings. The subject is actual and the paper is well written, but I do have some suggestions for the contents of the manuscript. To start with, there is some English bugs that should be corrected (for example, line 438; line 488; line 562; and so on).
1. How the authors could be sure that all excess of ionic liquid was removed from the membrane surface? The abbreviation for NP should be deciphered (line 63).
2. The zoom of Fig. B1 could be useful for clarity.
3. In Table 2 only tensile strength values are present (no elongation at break values).
4. In order to compare correctly the FTIR intensity, the spectra should be normalized. No information about it can be found in Experimental part.
5. The authors mentioned that the “the results can be related to an improvement in the interactions in the interphase IL-MOF-polymer” (lines 648-649). Which kind of interactions is present? Some explanation should be given.

Author Response
Dear Reviewer,
Please find in the document attach the response to your comments.
Kind regards,
Luísa Neves

Reviewer 2 Report
- Utilization of enhancement factor (EF) is not a suitable tool on effective evaluation on membrane performance. The utilization of filler enhancement index (Findex) should be used instead.
- In Table 2 (it should be Table C2), a reference value that is used to calculate the enhancement factor (EF) should be given.
- What is the reason that the author uses polysulfone, out of the other available polymers?
- Figure 5 is too difficult to see. Place important figures in Figure 5 and put other information into the supplementary information.
- Where is the solubility/diffusivity analysis?
- I do not agree on the use of 4th order polynomial to fit the isotherm. In thermodynamic standpoint, it is incorrect. It is impossible for gas adsorption analysis follows 4th order isotherm. The authors should use appropriate fitting equations. The authors should not just fit using mathematical expressions that gives the highest accuracy. Please amend accordingly.
- There is no error bar in Table 3.
- For Table C1 and Figure 7, there are no explanations on how the error bars are calculated.
- For the sorption selectivity, it is calculated as 50/50. However, for the gas permeation, it is determined at 15/85 (CO2/N2) and 40/60 (CO2/CH4). There is no consistency observed in this manuscript.
- No description in the experimental procedures on how the tensile test is performed.
Author Response

(The authors gave the same response as above.)

Reviewer 3 Report
a) I suggest to consider the size of the IR spectra to improve their "readability".
b) The choice of the "low" and "high" concentration of ionic liquids is clear. Nevertheless, their quantification (ca 3 and 30 wt.%) should be explained in more detail.
c) Although ionic liquids are relatively stable, maintaining a constant amount in the membrane can be problematic.
Author Response

(The authors gave the same response as above.)

Round 2
Reviewer 2 Report
Accepted at present form.